# Assessing the Effect of Cold Plasma on the Softening of Postharvest Blueberries through Reactive Oxygen Species Metabolism Using Transcriptomic Analysis

**DOI:** 10.3390/foods13071132

**Published:** 2024-04-08

**Authors:** Can Zhang, Jun-Hu Cheng

**Affiliations:** 1School of Food Science and Engineering, South China University of Technology, Guangzhou 510641, China; sole19760517@163.com; 2Academy of Contemporary Food Engineering, South China University of Technology, Guangzhou Higher Education Mega Centre, Guangzhou 510006, China; 3Guangdong Province Engineering Laboratory for Intelligent Cold Chain Logistics Equipment for Agricultural Products, Engineering and Technological Research Centre of Guangdong Province on Intelligent Sensing and Process Control of Cold Chain Foods, Guangzhou Higher Education Mega Centre, Guangzhou 510006, China

**Keywords:** blueberry softening, cold plasma, ROS metabolism, transcriptome

## Abstract

The postharvest softening and corresponding quality deterioration of blueberry fruits are crucial factors that hinder long-distance sales and long-term storage. Cold plasma (CP) is an effective technology to solve this, but the specific mechanism of delaying fruit softening remains to be revealed. Here, this study found that CP significantly improved blueberry hardness. Physiological analysis showed that CP regulated the dynamic balance of reactive oxygen species (ROS) to maintain hardness by increasing antioxidant content and antioxidant enzyme activity, resulting in a 12.1% decrease in the H_2_O_2_ content. Transcriptome analysis revealed that CP inhibited the expression of cell wall degradation-related genes such as the pectin hydrolase gene and cellulase gene, but up-regulated the genes of the ROS-scavenging system. In addition, the resistance genes in the MAPK signaling pathway were also activated by CP in response to fruit ripening and softening and exhibited positive response characteristics. These results indicate that CP can effectively regulate the physiological characteristics of blueberries at a genetic level and delay the softening process, which is of great significance to the storage of blueberries.

## 1. Introduction

Blueberries (*Vaccinium* spp.) are popular all over the world due to their desirable taste and health-promoting effects. They contain numerous nutrients such as phenolic acids, flavanols and anthocyanins [1]. However, blueberries in postharvest storage are prone to quality deterioration, thus reducing their edible quality and commercial value, which is not conducive to long-term storage and cross-regional sales, and this phenomenon restricts its development in the postharvest preservation industry. This limitation makes it difficult to meet the increasing demands for blueberry supply. Therefore, it is critical to explore a reliable method to extend the shelf life of blueberries by inhibiting firmness loss. In this area, many researchers have made great efforts. Ethylene absorbent was adopted to treat blueberry fruits in cold storage and effectively inhibited the softening and loss of the total phenol content of blueberries [2]. In addition, the activity of cell wall-degrading enzymes and ethylene-related enzymes were inhibited to hinder ethylene synthesis, which was consistent with ethylene exacerbating blueberry softening by stimulating enzyme activity and promoting the expression of genes related to cell wall degradation [3]. Additionally, chemical methods such as ethanol vapor [4,5], putrescine [6] and exogenous anthocyanins [7] were also reported to delay the softening of blueberries. Meanwhile, physical processing technology was also introduced into the research field. Chitosan/thyme oil coating combined with UV-C (short wave ultraviolet irradiation) treatment was adopted to inhibit the activity of polygalacturonase, pectin methylesterase, cellulose and β-glucosidase by controlling the expression intensity of related enzyme genes and reducing the loss of polysaccharides in the blueberry cell wall, thus maintaining the integrity of blueberry cell wall tissue structure and inhibiting the postharvest softening and senescence of blueberries [8].

Many factors can lead to the postharvest deterioration of blueberries, including ROS burst, mechanical damage and pathogenic bacteria infestation. ROS are produced normally after fruit harvesting, and the generated ROS are continuously scavenged due to the joint action of ROS-scavenging enzymes and endogenous antioxidant substances; thus, the generation and elimination of ROS are in a dynamic balance to prevent quality deterioration [9]. However, with the prolongation of postharvest storage, the balance tilts towards the accumulation of ROS, and excess ROS further reacts with lipids and proteins to damage the cell structure. Numerous studies have also shown that the dysregulation of ROS metabolism has a positive impact on accelerating the fruit softening process [10]. For instance, Chen et al. (2019) reported that the acidic electrolyzed oxidizing water treatment of blueberries not only significantly increased the activities of ROS-scavenging enzymes, including superoxide dismutase (SOD), catalase (CAT) and ascorbate peroxidase (APX), but also reduced the generation of superoxide anion (O_2_^-^) and cell membrane permeability to preserve blueberries [11]. Ge et al. (2018) also reported that γ-aminobutyric acid treatment delayed blueberry softening by regulating the phenylpropanoid pathway and ROS metabolism [12]. Therefore, it is of importance to investigate the relationship between the postharvest softening and ROS metabolism of blueberries during storage.

Cold plasma (CP) has emerged as an effective and promising processing technology in the food industry, with the advantages of environmental protection, low energy consumption, short processing time and easy operation [13,14]. It is an ionized gas consisting of oxygen radicals, charged particles, UV radiation, ozone and other reactive species. When partially oxygenated gases such as air or oxygen mixtures are discharged, oxygen free radicals, ozone and other reactive oxygen substances can exist stably, and they are also regarded as the main functional substances for food sterilization and preservation [15]. CP is increasingly applied in fruit and vegetable preservation, which has been widely used for bacterial inactivation on various kinds of fruits [16,17]. For example, Ji et al. (2020) showed that the appropriate application of CP enhanced the antioxidant system and reduced the accumulation of ROS (H_2_O_2_ and O_2_^−^) after 40 d storage of blueberries to achieve better postharvest quality [18]. Numerous studies suggested that CP could preserve fruit quality by affecting ROS metabolism.

In recent years, transcriptomic, metabolomic and proteomic technologies have been widely used in various physiological and biochemical studies of fruits, and transcriptomic sequencing is the fastest developing and most widely used sequencing method. Transcriptome sequencing (RNA-seq) is a technology that uses a high-throughput sequencing method to sequence all or part of mRNA, small RNA and non-coding RNA in cells or tissues. The most common transcriptome sequencing is based on second-generation sequencing technology, with lllumina’s NGS sequencing platform representing the mainstream. This method needs to process the RNA samples according to the experimental purpose, reverse transcribe part or all of the mRNA, miRNA and lncRNA into cDNA libraries, and then, sequence them using a high-throughput sequencing platform [19]. Usually, according to the different lengths of the sequenced objects, libraries of different sizes of fragments will be selected when the sequencing is built. In general, when mRNA sequencing is performed, libraries of fragments with sizes of several hundred base pairs are usually established during library construction, and bidirectional sequencing is often selected. When miRNA sequencing is performed, the miRNA is usually isolated and a small fragment library is established separately before one-way sequencing is performed. Long non-coding RNAs (lncRNAs) have forward and reverse transcription functions, so chain-specific library sequencing is often used [20]. To date, numerous studies have used transcriptomic techniques to investigate the changes in gene expression during fruit development and postharvest softening. Chen et al. (2017) revealed that nitric oxide (NO) delayed the cottony softening of wax apple via the regulation of cell wall degradation, carbohydrate metabolism, oxidation–reduction balance and plant hormone signal transduction pathways through transcriptome [21]. Ma et al. (2020) used a transcriptome to analyze the firmness of table grapes at three developmental stages and identified a lot of differentially expressed genes (DEGs) related to fruit firmness, such as those encoding PE (pectinesterase), PL (Pectinlyase), PG (Polygalacturonase) and XTH (Xyloglucan endotransglycosylase/hydrolase) [22].

Currently, few studies are available that use omics techniques to explore the mechanism of retarding of blueberry softening using CP treatment. Therefore, in this study, under the conditions of 25 °C and 85% relative humidity (RH), blueberries were treated with an appropriate dose of plasma, which directly demonstrated the effect of CP treatment on blueberry quality from the apparent characteristics, and we further analyzed the regulatory role of CP at the biomolecular level from the physiological aspects. Transcriptomics was also used to explore the gene and pathway regulation mechanisms of CP treatment affecting blueberry softening. This provides a reference for follow-up research on the preservation and storage of blueberries. The mechanism of the treatment of blueberries with cold plasma is shown in Figure 1.

## 2. Materials and Methods

### 2.1. Reagents and Chemicals

Phosphate-buffered saline (PBS) (0.1 M, pH 7.2) was bought from Beijing Oka Biological Technology Co., Ltd. (Beijing, China). Glacial acetic acid and absolute ethanol were purchased from Guangzhou Chemical Reagent Factory (Guangzhou, China).

### 2.2. Blueberry Samples

Blueberries (*Vaccinium* spp. cv. Emerald) during the commercially mature stage (approximately 80% maturity, approximately 3.5 N firmness) [7] were transported from a local blueberry plantation (Guangzhou, China) to our laboratory within two hours and stored in a refrigerator (BCD-525WKPZM (E), Hefei Midea Refrigerator Co., Ltd., Hefei, China) at 4 °C for 24 h to reduce the respiration rate. Blueberries with uniform size, no pests and diseases and no visible mechanical damage were selected as the experimental objects.

### 2.3. CP Treatment

The equipment used for sample processing (CPCS series, Nanjing Su Man Electronics Co., Ltd., Nanjing, China) belongs to the high-voltage electric field low-temperature plasma cold sterilization experimental system. The external size of the equipment is 1400 mm wide × 850 mm deep × 1200 mm high, and the overall color is stainless steel metal. The equipment structure is shown in Figure 2. The main function is to automate the experimental operation of food samples according to the preset parameters of the experiment, and record the relevant data. The plasma module is the core component of the equipment and is the actual area of the cold plasma experiment. This area can not only generate a high-voltage electric field during operation, but also change the size of the discharge interval for different samples. The plasma module structure is shown in Figure 3.

Approximately 25 g blueberries each time was placed in a polyethylene terephthalate (PET) tray (45 mm × 45 mm × 35 mm, 0.4 mm thickness), sealed with polyethylene (PE) film of 0.30 mm thickness, and then, placed between two rectangular copper plate electrodes (170 mm × 120 mm) of a high-voltage electric field dielectric barrier discharge (DBD) set-up (Nanjing Suman Plasma Technology Co., Ltd., Nanjing, China). Plasma was produced by ionized air in atmospheric conditions (20 °C and 65% RH). The processing distance between the two electrodes was about 40 mm. In the experiment, blueberries were exposed to 100 kV for 6 min (interval discharge, 30 s discharge, 10 s rest) with a frequency of 50 Hz, defined as CP-6, and then, stored at 25 ± 0.5 °C and 85% RH for 8 d. And the operating power of the DBD was 5.0 kW. Samples were collected every other day for determination.

### 2.4. Determination of Emission Spectrum

Emission spectrometers were used to identify the active components of cold plasma discharge in an atmospheric environment. The optical fiber diameter was 0.22 μm, and the spectral range was from 200 nm to 1100 nm. During the discharge process, the probe was penetrated into the gap between the two electrode dielectric plates and was 1.5 cm away from the electrode in the horizontal direction. The active substances of cold plasma discharge in the air were analyzed based on the spectrogram [23].

### 2.5. Firmness Test

Firmness was determined by a texture analyzer (TA. XTPlusC, Stable Micro System, Sidcup, UK) according to a previous method [24], with little modification. Fifteen blueberries of each group were assayed, and the equatorial portion of each blueberry was punctured symmetrically twice to a depth of 8 mm at rate of 1.0 mm/s with a P/2 stainless steel probe (Φ2 mm). The maximum force of each test was recorded and expressed in Newtons (N). A schematic diagram of the determination of blueberry firmness by texture analyzer is shown below in Figure 4.

### 2.6. H_2_O_2_ Content

H_2_O_2_ content was determined using assay kits (Nanjing Jiancheng Biotechnology Co., Ltd., Nanjing, China) [25]. A total of 5 g blueberry tissue was added to 45 mL 0.1 M PBS (pH 7.2), homogenized in an ice water bath and centrifuged at 10,000 rpm for 10 min (4 °C). We used 10% of the supernatant for further testing according to manufacturer’s instructions.

### 2.7. Antioxidant Metabolism

#### 2.7.1. Antioxidant Content

Ascorbic acid (AsA) and glutathione (GSH) contents were measured by a colorimetric method using assay kits (Nanjing Jiancheng Biotechnology Co., Ltd., Nanjing, China) with absorbance of 536 and 420 nm, respectively, which were expressed as μg mgprot^−1^ and gGSH gprot^−1^, respectively. A total of 5 g blueberry tissue each time was weighed for the preparation of the supernatant.

#### 2.7.2. Antioxidant Enzyme Activity

SOD and POD activities were determined with assay kits (Shanghai Yuanye Biotechnology Co., Ltd., Shanghai, China) at absorbance levels of 550 and 470 nm, respectively, and CAT and APX activities were determined, respectively, at absorbance levels of 405 and 290 nm using assay kits (Nanjing Jiancheng Biotechnology Co., Ltd., Nanjing, China). Blueberry tissue was homogenized with extraction buffers in an ice water bath at a ratio of 1:2 (POD), 1:4 (SOD) and 1:9 (CAT and APX), respectively, and then, centrifuged (4 °C) to obtain the supernatant for the enzyme activity assay. SOD activity was expressed as U g^−1^ fresh weight (FW), and 1 U is the amount of SOD corresponding to 50% inhibition of enzyme per gram of tissue in 1 mL of reaction system; CAT activity was expressed as U mgprot^−1^, and 1 U is defined as 1 μmol of H_2_O_2_ per milligram of tissue decomposed per minute; POD activity was expressed as U g^−1^ FW, and 1 U is the amount of enzyme required for a 0.01 change in absorbance per minute; APX activity was expressed as U g^−1^ FW, and per gram of tissue, catalyzes 1 μmol of AsA per minute per milliliter of reaction system, defined as 1 U.

#### 2.7.3. Total Antioxidant Capacity (T-AOC) Determination

A total of 5 g blueberry tissue was homogenized with 45 mL 0.1 M PBS (pH 7.2) in an ice water bath, and then, centrifuged (2500 rpm, 4 °C) for 10 min to obtain the supernatant for further assay. The determination of T-AOC was carried out by means of assay kits (Nanjing Jiancheng Biotechnology Co., Ltd., Nanjing, China) at an absorbance of 520 nm, which was expressed as U mgprot^−1^, and 1 U is defined as a 0.01 increase in the absorbance value of the reaction system per milligram of tissue protein per minute [26].

### 2.8. Transcriptomic Analysis

#### 2.8.1. Transcriptome Sequencing and Gene Expression Analysis

The above results showed that during the period of 0 to 8 days, cold plasma treatment could delay the softening process of blueberry fruits, and most of the indexes changed significantly on the second day of storage. Therefore, samples stored at 0, 2 and 8 d were selected as omics sampling points, which was carried out by Shanghai Biotree Biomedical Technology Co., Ltd., and three independent biological replicates were made of each sample for a total of 18 samples named CK01, CK02, CK03, CK21, CK22, CK23, CK81, CK82, CK83, CP01, CP02, CP03, CP21, CP22, CP23, CP81, CP82 and CP83. Raw reads obtained after sequencing were stored in FASTQ file format, which cannot be used for subsequent bioinformation analysis. Clean reads were obtained from the removal of unqualified sequencing reads in raw reads, and then, aligned to the blueberry reference genome using the mapping tool Hisat2. Differential expression analysis of genes between two samples was performed with the DESeq R package (1.18.0), and the DEGs met the following criteria: adjusted *p*-value ≤ 0.05 and |log2Fold Change| ≥ 1.

#### 2.8.2. DEG Enrichment Analysis

Enrichment analysis of the DEGs based on both the Kyoto Encyclopedia of Genes and genomes (KEGG) and Gene Ontology (GO) databases was performed. After enrichment analysis, KOBAS 3.0 software was used to conduct statistical analysis of the DEGs in the pathways of GO and KEGG.

### 2.9. Statistical Analysis

IBM SPSS Statistics 25.0 (IBM Co., Armonk, NY, USA) was applied to analyze the experimental data, and the significance of the differences was tested using Duncan’s multiple comparison method, with *p* < 0.05 indicating significant differences. Moreover, Origin 8.5 software (OriginLab Co., Northampton, MA, USA) was used for plotting. Three replicates were measured for each experiment to ensure data accuracy.

## 3. Results and Discussion

### 3.1. Emission Spectroscopic Determination of Plasma

Optical emission spectroscopy can be used to qualitatively analyze active plasma particles. The qualitative detection of ROS helps to provide an understanding of the mechanism of action that retards the softening of blueberries. The emission spectroscopy of plasma discharge in air is shown in Figure 5. Specifically, as can be seen from the figure, the strong peaks mainly belonged to nitrogen-containing particles. ROS species were also observed, with weak peaks of O_3_ at 299 nm and 330–390 nm, ∙OH at 319 nm and OH^+^ at 407 nm [27,28,29]. It can be seen from the figure that the emission lines related to oxygen were limited, on the one hand, due to the high reactivity of oxygen-containing active substances; on the other hand, it may be due to the quenching and consumption of O (3P) and O (5P) in air [30].

### 3.2. Firmness Changes

The effect of CP treatment on blueberry firmness is shown in Figure 6A. Blueberry firmness gradually decreased as the storage time increased, with the firmness of CP-treated blueberries being significantly higher than that of the control (*p* < 0.05). The fruit firmness of the CP-treated and control groups decreased to 1.45 N and 1.18 N, respectively, at the end of storage, indicating that CP treatment obviously retarded the firmness loss of postharvest blueberries. This result is consistent with previous studies which showed that CP treatment could maintain higher firmness in fresh-cut strawberries [31], grapes [32] and apples [33].

### 3.3. H_2_O_2_ Content Changes

As a kind of long-lived ROS, H_2_O_2_ accumulation in postharvest fruit can damage cell membranes, cause metabolic activity disorder and even lead to cell death [34]. During the early storage period (0–2 d), H_2_O_2_ accumulation was significantly induced by CP (*p* < 0.05), which was probably due to the fact that CP treatment promoted a surge in H_2_O_2_ in the early stage and, thus, activated the resistance response of blueberries (Figure 6B). Afterwards, a rapid increase in H_2_O_2_ content in the control group was observed, while its rise in the CP-treated group was more moderate. This could be explained by the activation of the antioxidant system induced by UV irradiation and ozone in CP, thus accelerating the removal of ROS [35,36].

### 3.4. The ROS-Scavenging System

The generation and scavenging of ROS are reported to have essential impacts on fruit ripening and senescence. The AsA-GSH cycle is a crucial antioxidant system in which two key non-enzymatic antioxidants (AsA and GSH) are able to scavenge excess ROS and maintain redox homeostasis in cells to delay fruit senescence. As shown in Figure 7, the GSH and AsA contents in the control and CP-treated blueberries exhibited an increase first, and then, a decrease with the extension of storage, and both of them reached their peak at 2 d. The GSH content was significantly higher in the CP-6 group than in the control from 4 d (*p* = 0.022 < 0.05), as shown in Figure 7A, while CP treatment had similar effect on AsA content from 2 d, as shown in Figure 7B, which reveales that CP treatment improved the efficiency of the AsA-GSH cycle.

Antioxidant enzymes such as SOD, CAT, POD and APX also play effective roles in regulating ROS metabolism during fruit senescence. SOD catalyzes O_2_^−^ to O_2_ and H_2_O_2_, which is considered to be the first line of defense against the production of the toxic effects of O_2_^−^. CAT, POD and APX further converted H_2_O_2_ to H_2_O and O_2_ to protect cells from the damage of H_2_O_2_. In Figure 8A, it is shown that SOD activity increased to the maximum at 2 d, and then, decreased. The activity of SOD was 10.9% higher in the CP-6 group than in the control at 2 d, and CP-treated blueberries exhibited significantly higher SOD activity during storage (*p* = 0.024 < 0.05). A similar change is also observed in the activities of POD and APX in Figure 8C,D, respectively, which show that CP significantly inhibited the decline of the activities of POD and APX from 2 d (*p* = 0.016 < 0.05, *p* = 0.009 < 0.05). Compared to the control fruits, the activities of POD and APX increased by 30.8% and 61.1%, respectively, at 8 d after CP treatment. Conversely, in Figure 8B, it is shown that the CAT activity of postharvest blueberries decreased first in two groups, while it decreased significantly faster in the control group (*p* = 0.039 < 0.05). At the end of storage, CAT activity in treatment group was 1.14 times higher than that in the control group. Many studies have shown that CP affects the antioxidant enzyme activity of fruits, as well. Dong and Yang. (2019) found that SOD activity in blueberries showed the maximum increase of 79.3% following different CP treatments compared with the control [37]. It was also reported that the SOD activity of button mushrooms was increased by 23.3%, 25.1% and 47.3%, respectively, compared to untreated samples when immersed in plasma-activated water for 5, 10 and 15 min after 7 d of storage [38,39]. Similarly, the activities of SOD, CAT, POD and APX in blueberries were obviously increased after CP treatment [40].

As shown in Figure 9, the T-AOC increased from 0 to 4 d, and then, decreased in both the CP-treated and control blueberries, and CP treatment significantly enhanced T-AOC during the later storage (*p* = 0.015 < 0.05). The T-AOC of the CP-6 group was 7.77 U gprot^−1^, obviously higher than that of the control at 6.42 gprot^−1^ at 8 d, which demonstrates a relationship with the increasing content of GSH and AsA (Figure 7) and the activities of multiple enzymes (Figure 8). Previous studies also showed that CP treatment had a positive effect on the T-AOC of postharvest blueberries [18,41].

According to the analysis of the above indicators, CP treatment effectively maintained the firmness of blueberries and kept the H_2_O_2_ content at a low level in the late storage period (after 4 d). Meanwhile, the efficiency of the AsA-GSH antioxidant cycle system and the main antioxidant enzyme activities of blueberries in the CP treatment group were significantly higher than those in the control group in the late storage period. This was consistent with the decreasing trend of H_2_O_2_ content in the late storage period, and the measured total antioxidant capacity also showed the same trend.

### 3.5. Transcriptome Data and Differential Gene Expression

The above analysis showed that CP treatment inhibited firmness loss and regulated ROS metabolism in blueberries, with most indicators exhibiting a significant difference from 2 d. Therefore, to better understand the mechanism of CP in delaying blueberry softening, this study sequenced blueberries from the treatment and control groups at 0, 2 and 8 d of storage.

As shown in Figure 10A, 552, 467 and 425 DEGs were detected in the samples stored at 0, 2 and 8 d, respectively, and the number of down-regulated DEGs was more than that of up-regulated genes in all groups. In the CP0d-vs-CK0d samples, 197 were up-regulated and 355 were down-regulated; in the CP2d-vs-CK2d samples, 161 were up-regulated and 306 were down-regulated; and in the CP8d-vs-CK8d samples, 95 were up-regulated and 330 were down-regulated. Figure 10B shows that 310 identical DEGs were detected between 0 d and 2 d, while 108 identical DEGs were detected between 0 d and 8 d and 116 identical DEGs were detected between 2 d and 8 d, which indicates that the effects of CP treatment on blueberry fruit in the early storage period (0 and 2 d) were relatively similar, while the effects on blueberry fruit in the later storage period (8 d) were much different to those in the early stage.

### 3.6. Key DEG Analysis

#### 3.6.1. Dentification and Analysis of DEGs Involved in the Cell Wall Metabolism Pathway

A total of 15 DEGs involved in cell wall metabolism were identified to study the effects of CP on postharvest blueberries during storage. As shown in Table 1, after CP treatment, four genes encoding pectin hydrolase, including PG, PE, β-galactosidase (β-Gal) and PL, were down-regulated, and *PG*, *PE* and *BGAL* were significantly down-regulated during the whole storage period, while *PL* was most significantly down-regulated at 2 d, and the specific gene expression level expressed by the TPM (transcripts per million) index is shown in Table 2. The DEGs mainly involved in cellulose degradation were composed of three genes encoding XTH, one encoding β-glucosidase (GLU) and two encoding β-1,3-endoglucanase, with three DEGs exhibiting significantly down-regulated expression, and the other three showing slightly up-regulated expression. In addition, one gene was found to be closely related to expansin, which modifies the cell wall as a result of cell swelling, leading to fruit softening, and the gene *EXPA8* of blueberries was significantly down-regulated after CP treatment at 8 d. The remaining four DEGs were involved in starch metabolism, with *BAM3*, encoding β-amylase, slightly up-regulated at 0 d; *AMY1.1*, encoding α-amylase, down-regulated at 8 d; and *SUS1* and *WAXY*, encoding sucrose synthase, both significantly down-regulated during storage.

Plant cell walls are dynamic substrates for polysaccharide networks, and the degradation of cell wall polysaccharide fractions is thought to be an important factor causing fruit softening [42,43]. During fruit ripening, the cell wall undergoes multiple chemical reactions simultaneously, including synthesis, crosslinking and hydrolysis, which involve many proteins and enzymes [44]. In the transcriptomic analysis of the softening process in two kinds of table grapes, Ma et al. (2020) found that the differential expression of cell wall modification-related genes such as *PE*, *PL*, *PG*, B*GAL*, *GATL*, *WAK*, *XTH* and *EXP* might be responsible for the differences in firmness between “Red Globe” and “Muscat Hamburg” [22]. A similar discovery was also observed in sweet cherry softening with genome-wide identification [45]. In this study, significantly down-regulated expression was observed in fifteen DEGs involved in cell wall degradation, which further confirmed the CP inhibition of the decline in blueberry firmness Figure 6A. The above results show that CP treatment significantly down-regulated the expression of genes related to cell wall degradation and hindered the metabolism of cell wall components to maintain blueberry firmness.

#### 3.6.2. Dentification and Analysis of DEGs Involved in the ROS Metabolism Pathway

As fruit softening occurs, the accumulation of ROS is augmented and the activities of ROS-scavenging enzymes is attenuated, which results in oxidative damage and even the death of cells. As shown in Table 3, after CP treatment, a total of twelve DEGs in the samples were found to participate in ROS metabolism, including *CAT1* and *CAT2* encoding peroxidase, *SODCC*, *SODCC.3* and *SODCP* encoding superoxide dismutase, *GSVIVT00023967001* encoding peroxidase, *APX1* encoding L-ascorbic peroxidase, *GSTL3* encoding glutathione S-transferase, *GSTU20*, *DHAR3* and *GSTU19*, and *GPX7* encoding glutathione peroxidase. The expression of these DEGs was generally up-regulated, except for *GSTL3*, which was down-regulated at 8 d. Some of DEGs, including *SODCC*, *SODCP*, *GPX7*, *GSTU19* and *DHAR3*, were only significantly up-regulated at 8 d. These results exhibit that CP treatment induced the up-regulated expression of antioxidant enzyme genes and promoted the AsA-GSH cycle, thus protecting blueberries from oxidative damage to delay postharvest senescence.

As shown in Figure 6B, CP treatment significantly maintained the ROS homeostasis of blueberries, effectively improved antioxidant capacity and inhibited H_2_O_2_ accumulation. Through the analysis of DEGs, twelve DEGs encoding the antioxidant enzymes SOD, CAT, APX, POD, GST and GPX were significantly up-regulated, further confirming, at the molecular level, that CP could delay blueberry softening by promoting the expression of antioxidant system-related genes. Metabolomic analysis also revealed that CP treatment induced the expression of ROS-scavenging enzyme genes in fresh-cut strawberries, significantly augmented antioxidant levels and delayed the deterioration of strawberry quality [46]. Furthermore, there is solid evidence that CP treatment is beneficial in inducing phenolic accumulation and enhancing antioxidant activity in fresh-cut papaya fruit [47].

Common active substrates generated by CP are reactive oxygen/nitrogen species (RONS) and charged particles such as ozone, O_2_^−^, NO, hydroxyl radical (•OH), H_2_O_2_ and electrons [48,49]. It is reported that the presence of reactive species is closely related to cell wall-degrading enzymes, and it has been confirmed that these active substances in CP are capable of modifying protein amino acid residues and disrupting the spatial structure and active sites of enzymes [50]. In addition, the above-mentioned reactive species might also take part in regulating ROS signaling in fruit to inhibit the activity of cell wall-degrading enzymes. When fruits are subjected to CP-induced oxidative stress, the intracellular ROS in fruits will change accordingly, making ROS act as signaling molecules that regulate the biochemical reactions of the fruit body to counteract external stress [51]. In this process, ROS signaling may stimulate the cell wall metabolic system and regulate the activity of related enzymes in some way. The exact mechanism through which the CP-generated reactive species retard the softening of blueberries needs to be further investigated.

#### 3.6.3. Dentification and Analysis of DEGs Involved in the MAPK Signaling Pathway

The mitogen-activated protein kinase (MAPK) signaling pathway is ubiquitous in plants as a signaling module involved in regulating plant growth, development and programmed death, as well as in responses to various external stimuli (cold/hot/drought/ROS/irradiation/pathogenic microorganisms, etc.) [52]. In Table 4, it is shown that fourteen DEGs were associated with the MAPK signaling pathway, and they all responded positively to different kinds of stress. *CAT1* and *CAT2* encoding peroxidases and *PYL* encoding abscisic acid receptors were located in the pathway response to salt/drought/osmotic stress; six DEGs were located in the response to pathogenic stress, including *FLS2* encoding serine/threonine protein kinase; *NME2* and *NDPK2* encoding nucleoside diphosphate kinases; *WRPK7* and *WRPK75* encoding WRKY transcription factors; and *PR1* encoding pathogenesis-related protein 1. Among them, *FLS2* and *NDPK2* were only up-regulated at 0 d; *WRPK7*, *WRPK75* and *PR1* were only up-regulated at 8 d, and they were all related to the promotion of phytochemical synthesis and resistance to pathogen attacks; two DEGs were detected in response to phytohormone—*NPL7*, encoding transcription factor MYC2, which promotes both the defense response and damage response in fruit, was up-regulated at 0 d, and another gene encoding endo-chitinase, also associated with the defense response, was only down-regulated at 8 d; the remaining three DEGs encoded calmodulin, with *My16* up-regulated at 0 d, and *PCM5* and *CAM1-1* up-regulated at 8 d, and they all responded to damage and had a positive effect on retaining ROS homeostasis in fruit. The above results indicate that CP stimulation activated the MAPK signaling pathway, making fruit respond positively to multiple biotic and abiotic stresses to maintain good postharvest quality. In addition, the most significant effects were observed at the beginning (0 d) and end (8 d) of storage, probably due to the sudden stimulation by CP at 0 d and the increased softening effect on the fruit at 8 d.

ROS accumulation is the foremost biochemical response of plants to abiotic stress, such as oxidative stress produced by CP, and as a second messenger, ROS convey messages from extracellular stimuli to the intracellular space by activating various signaling pathways inside the cells, including the MAPK signaling pathway [53]. It is well known that the MAPK signaling pathway is closely related to ROS production and transformation. Many studies reported that exogenous H_2_O_2_ activated *MAPK3*, *MAPK6* and *MAPKKK* of the MAPK cascades in Arabidopsis thaliana [54,55]. Similarly, H_2_O_2_ was also confirmed to increase the expression of *NDPK2* to improve strength the tolerance against various stresses [56]. These results demonstrated that the MAPK signaling pathway is affected by ROS metabolism. In this study, we found that exogenous reactive species produced by CP might be responsible for the significant change in gene expression in the MAPK signaling pathway.

## 4. Conclusions

The mechanism of CP-delayed blueberry softening was explored through physiological and transcriptome analyses in this study. Significantly higher firmness of CP-treated blueberries was observed during storage. CP treatment retained ROS homeostasis by increasing antioxidant contents and inducing antioxidant enzyme activities to inhibit H_2_O_2_ accumulation. CP regulation was involved in multiple biological processes at the molecular level. CP treatment suppressed the expression of pectin hydrolase genes, cellulose hydrolase genes, expansin genes and starch-degradation-related genes, such as *PG*, *PL*, *PE*, *BGAL*, *GLU*, *XTH*, *EXPA8* and *AMY*, to inhibit cell wall degradation. Moreover, the up-regulation of ROS-scavenging-related genes, such as *CAT*, *SOD*, *POD*, *APX* and *GST*, may also be responsible for maintaining blueberry firmness, which further confirms the results of the previous physiological analysis of ROS metabolism. In addition, CP also affected the MAPK signaling pathway by enhancing its defense capacity against oxidative stress. In conclusion, CP-mediated ROS metabolism drove different biological responses to delay postharvest blueberry softening.

## Figures and Tables

**Figure 1 foods-13-01132-f001:**
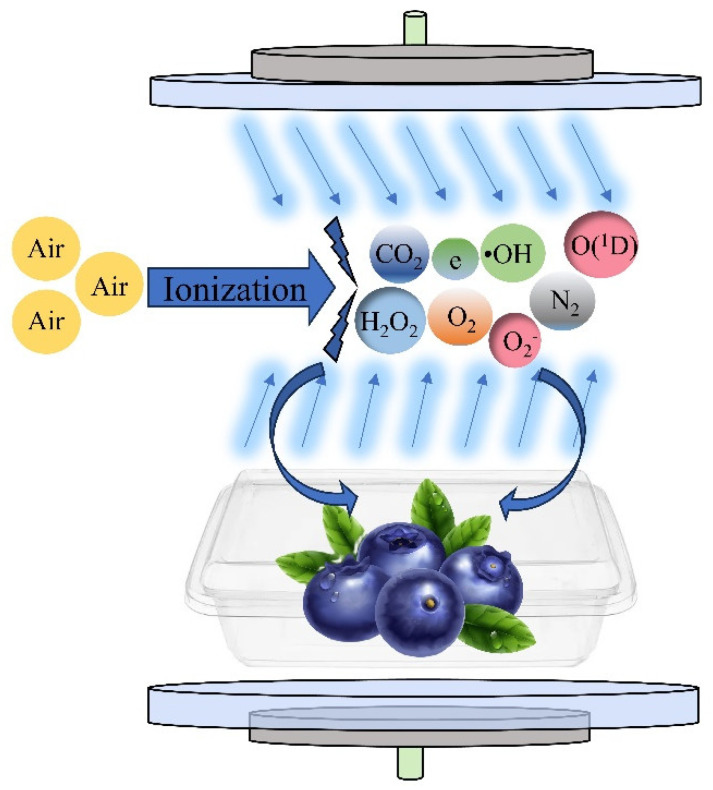
Mechanism of treatment of blueberries with cold plasma.

**Figure 2 foods-13-01132-f002:**
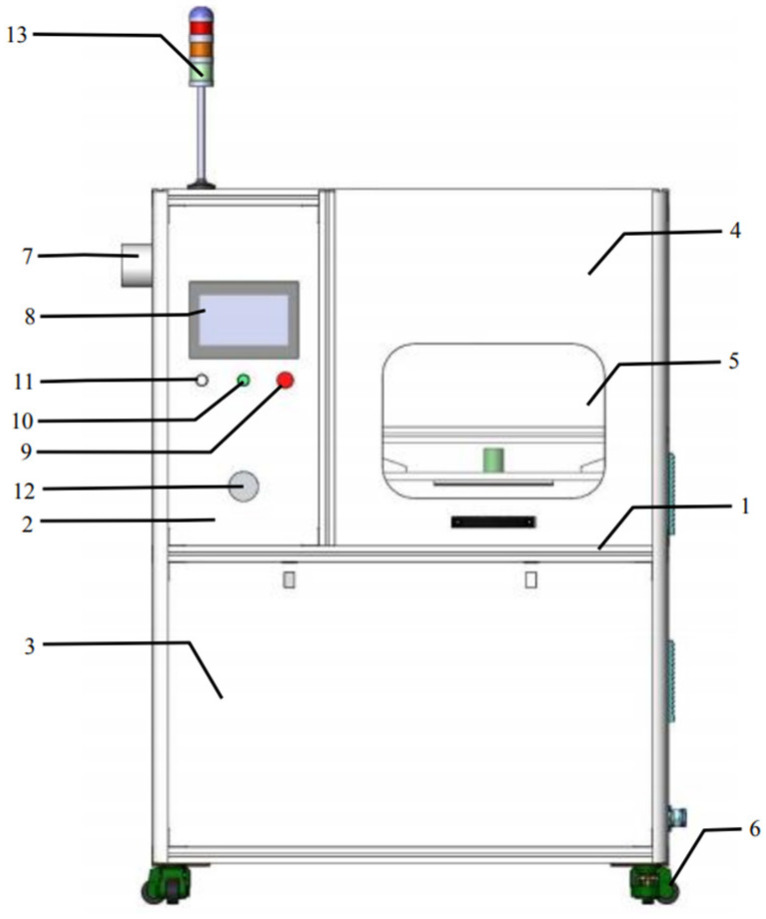
Depiction of the special high-voltage electric field CPCS cold sterilization experimental equipment. 1: Aluminum profile frame; 2: stainless steel panel; 3: stainless steel valve; 4: counterweight safety door; 5: observation window; 6: foot fuma wheel; 7: heat dissipation exhaust port; 8: touch screen; 9: emergency shutdown switch; 10: power-on button; 11: status indicator light; 12: voltage control knob; 13: tricolor alarm light.

**Figure 3 foods-13-01132-f003:**
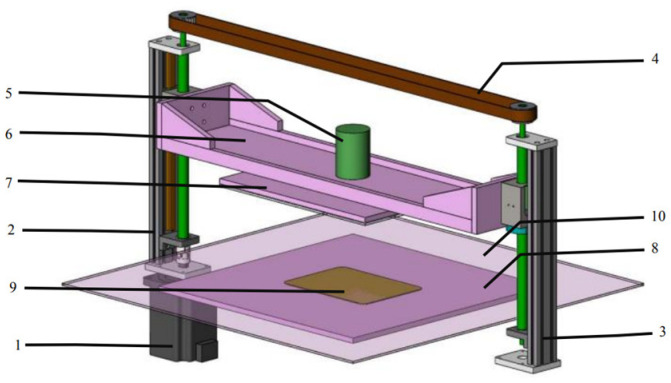
Plasma module structure. 1: servo motor; 2: active lead screw slide table; 3: passive lead screw slide table; 4: synchronous belt drive system; 5: high-voltage terminal block; 6: upper electrode beam; 7: upper high-voltage electrode; 8: lower electrode baseplate; 9: lower electrode plate; 10: lower electrode dielectric plate.

**Figure 4 foods-13-01132-f004:**
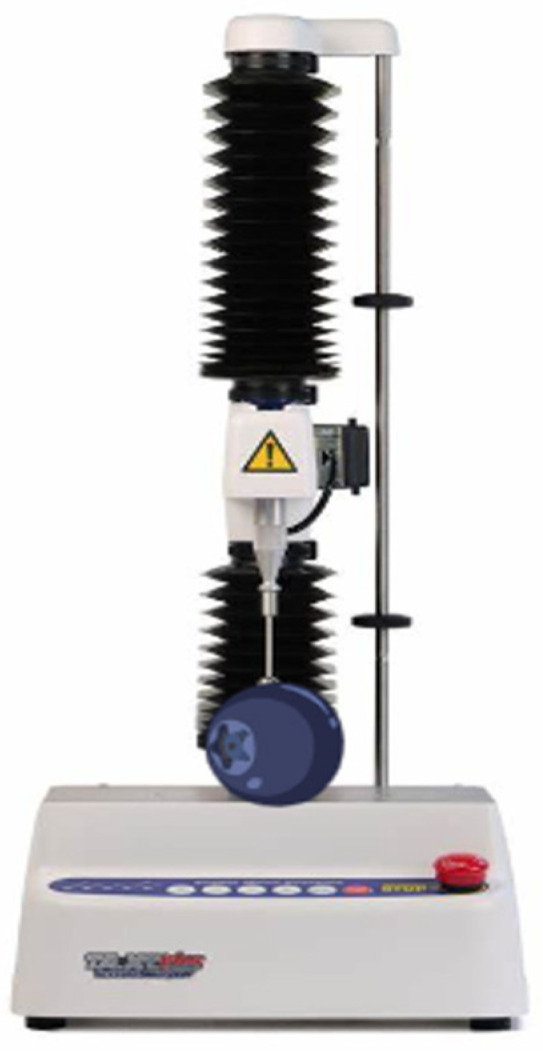
Schematic diagram of determination of blueberry firmness by texture analyzer.

**Figure 5 foods-13-01132-f005:**
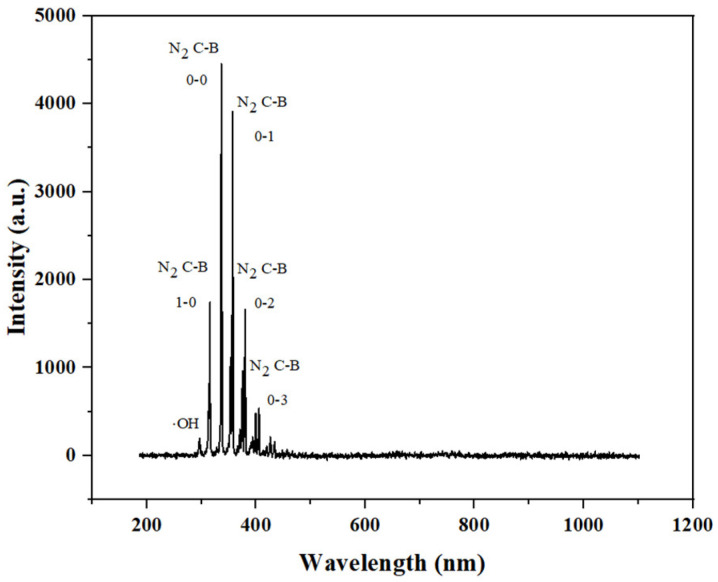
Emission spectroscopic determination of plasma.

**Figure 6 foods-13-01132-f006:**
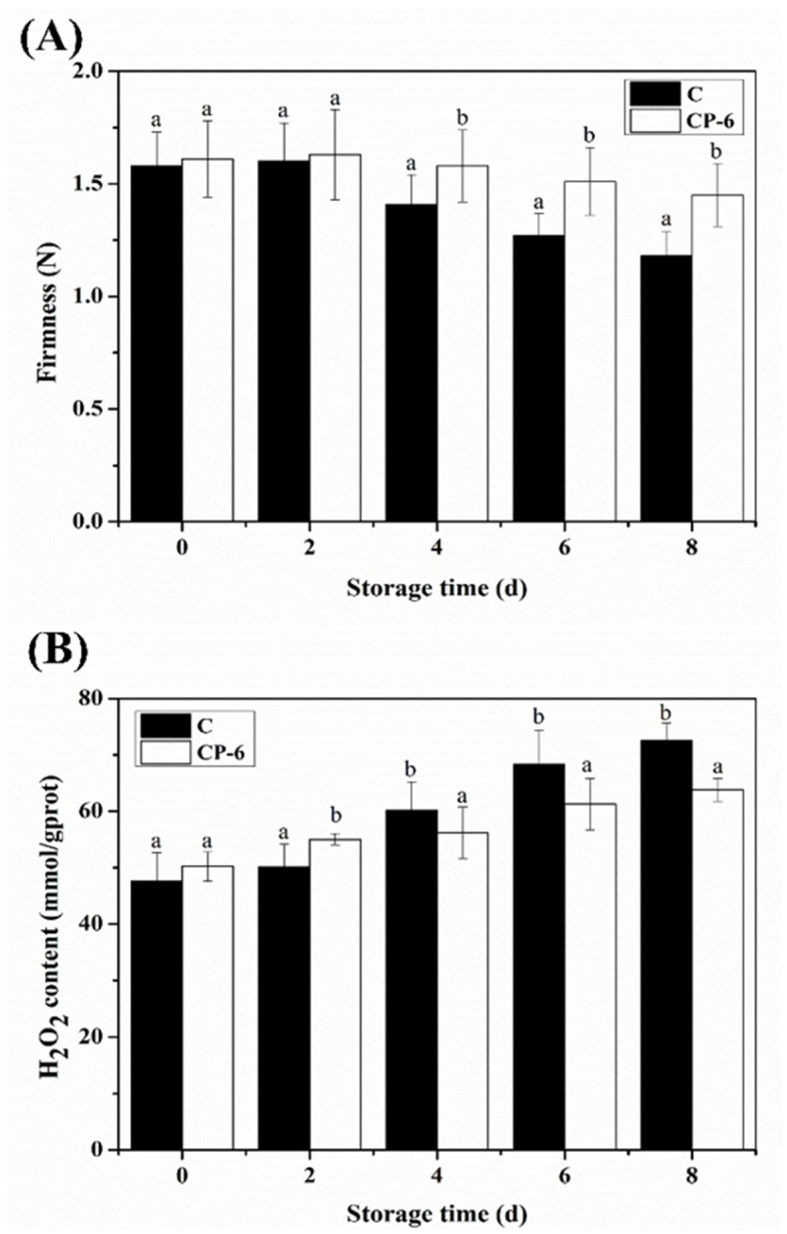
Firmness (**A**) and H_2_O_2_ content (**B**) of blueberry fruit treated with cold plasma during storage: C (control) and CP-6 (cold plasma treatment for 6 min). Means on the same storage day with different small letters are significantly different (*p* < 0.05).

**Figure 7 foods-13-01132-f007:**
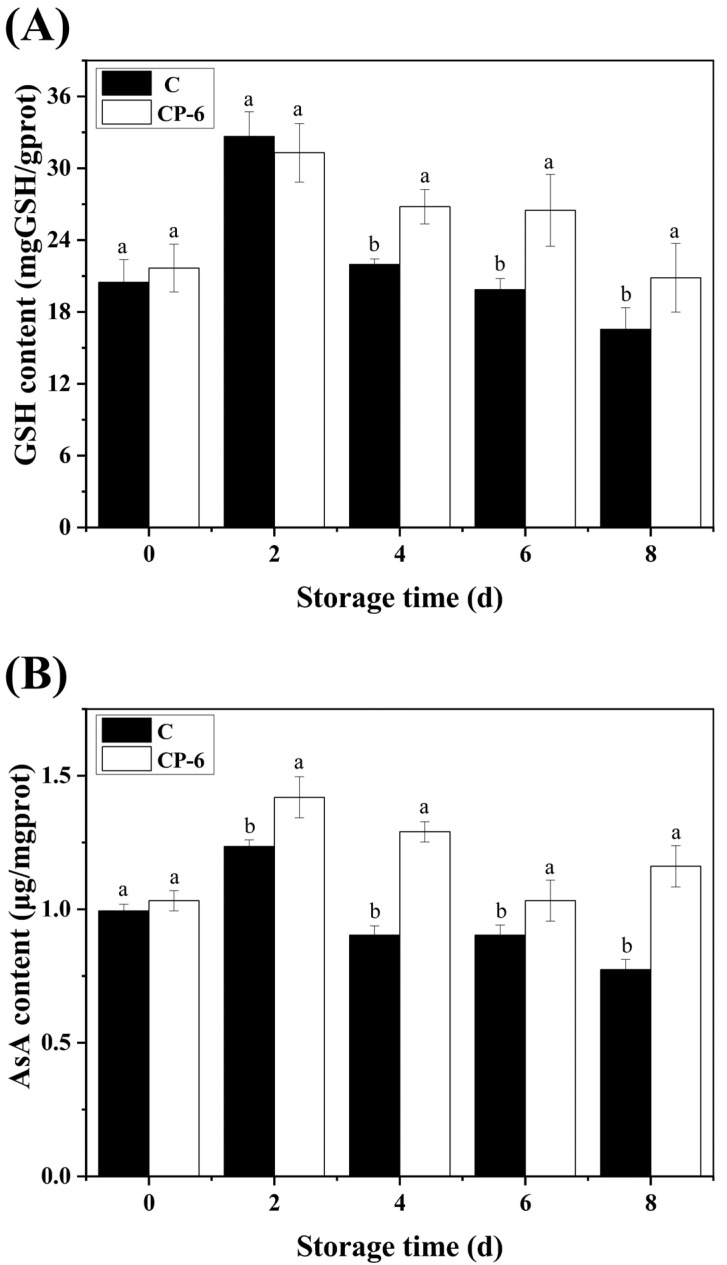
GSH (**A**) and AsA (**B**) content variation in blueberry fruit caused by cold plasma treatment: C (control) and CP-6 (cold plasma treatment for 6 min). Means on the same storage day with different small letters are significantly different (*p* < 0.05).

**Figure 8 foods-13-01132-f008:**
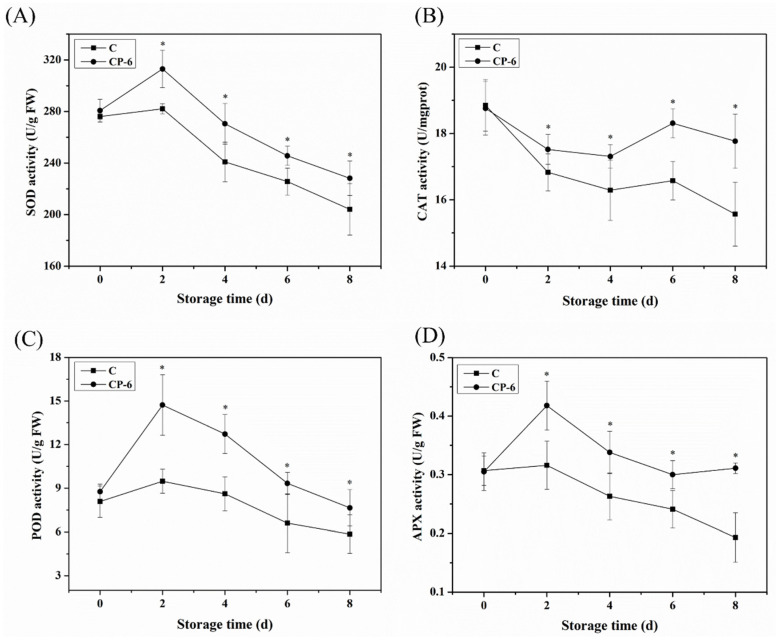
Effect of cold plasma treatment on SOD (**A**), CAT (**B**), POD (**C**) and APX (**D**) activity of blueberry fruit: C (control) and CP-6 (cold plasma treatment for 6 min). Means on the same storage day with “*” are significantly different (*p* < 0.05).

**Figure 9 foods-13-01132-f009:**
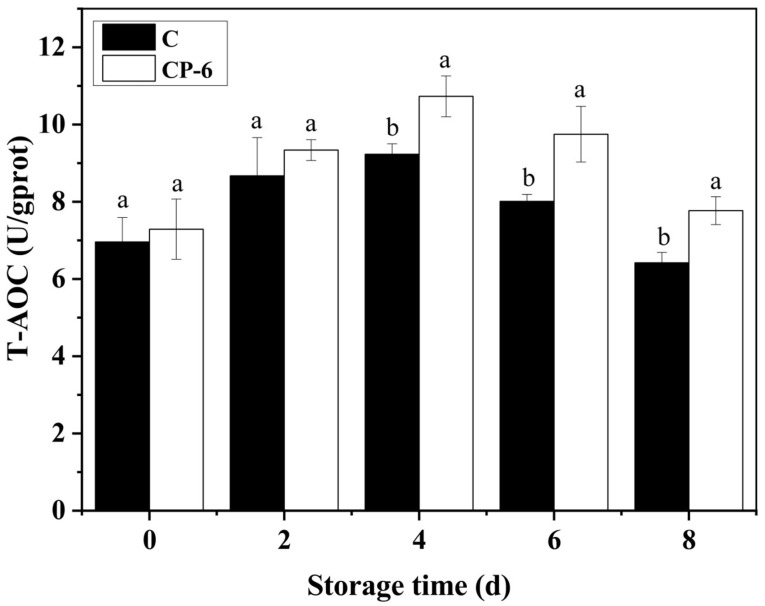
T-AOC content of blueberry fruit after cold plasma treatment: C (control) and CP-6 (cold plasma treatment for 6 min). Means on the same storage day with different small letters are significantly different (*p* < 0.05).

**Figure 10 foods-13-01132-f010:**
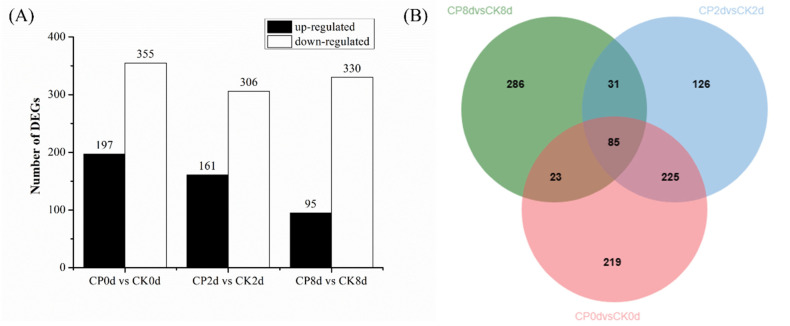
Histogram (**A**) and Venn diagram (**B**) of the statistics of the number of DEGs.

**Table 1 foods-13-01132-t001:** Expression of DEGs related to cell wall metabolism in blueberry fruit.

Gene ID	Name	Functional Annotation	log_2_Fold Change		
			CP0d-vs-CK0d	CP2d-vs-CK2d	CP8d-vs-CK8d
TRINITY_DN3072_c0_g1	PG	Polygalacturonase	−2.76	−6.84	−7.79
TRINITY_DN27616_c0_g3	PE	Pectinesterase	-	-	−2.97
TRINITY_DN32319_c1_g1	BGAL	Beta-galactosidase	-	−1.10	−3.89
TRINITY_DN39374_c2_g2	PL	Pectate lyase	-	−7.27	−4.48
TRINITY_DN17987_c1_g1	XTH8	Xyloglucan endotransglucosylase/ hydrolase protein 8	−7.20	-	-
TRINITY_DN18512_c0_g1	XTH8	Xyloglucan endotransglucosylase/ hydrolase protein 8	-	−7.06	−4.99
TRINITY_DN33049_c0_g4	XTH23	Xyloglucan endotransglucosylase/ hydrolase protein 23	1.10	-	2.02
TRINITY_DN36105_c0_g1	GLU	Beta-glucosidase	−9.77	−9.80	-
TRINITY_DN34957_c0_g2	At4g29360	Beta-1,3-endoglucanase 12	-	1.51	1.11
TRINITY_DN28630_c2_g1	At1g32860	Beta-1,3-endoglucanase 11	-	-	1.28
TRINITY_DN11208_c0_g1	EXPA8	Expansin	-	-	−7.22
TRINITY_DN22419_c0_g2	BAM3	Beta-amylase 6	1.11	-	-
TRINITY_DN28798_c0_g2	SUS1	Sucrose synthase	−10.07	-	-
TRINITY_DN8373_c0_g1	WAXY	Granule-bound starch synthase 1	−8.41	−7.71	−5.08
TRINITY_DN34835_c0_g1	AMY1.1	Alpha-amylase	-	-	−1.44

Note: “-” in this table means that the gene was not significantly expressed (*p* > 0.05).

**Table 2 foods-13-01132-t002:** Gene expression levels of PG, PE, β-galactosidase (β-Gal) and PL.

Gene ID	Name	Functional Annotation	Gene Expression Level Expressed by TPM					
			CP0d	CK0d	CP2d	CK2d	CP8d	CK8d
TRINITY_DN3072_c0_g1	PG	Polygalacturonase	0.27	0.91	0.00	0.91	0.00	3.80
TRINITY_DN27616_c0_g3	PE	Pectinesterase	2.54	1.87	0.94	1.19	0.26	2.53
TRINITY_DN32319_c1_g1	BGAL	Beta-galactosidase	3.79	3.44	34.64	77.51	0.33	6.63
TRINITY_DN39374_c2_g2	PL	Pectate lyase	618.82	922.6	0.00	2.24	0.25	7.84

**Table 3 foods-13-01132-t003:** Expression of DEGs related to ROS metabolism in blueberry fruit.

Gene ID	Name	Functional Annotation	log_2_Fold Change		
			CP0d-vs-CK0d	CP2d-vs-CK2d	CP8d-vs-CK8d
TRINITY_DN41456_c0_g1	CAT1	Catalase isozyme 1	−1.71	6.18	-
TRINITY_DN12758_c0_g1	CAT2	Catalase isozyme 2	1.19	5.95	1.55
TRINITY_DN976_c0_g1	SODCC	Superoxide dismutase	-	-	2.51
TRINITY_DN18708_c1_g1	SODCC.3	Superoxide dismutase [Cu-Zn]	1.09	1.86	-
TRINITY_DN18363_c0_g1	SODCP	Superoxide dismutase [Cu-Zn]	-	-	3.21
TRINITY_DN22770_c1_g1	POD	Peroxidase 4	1.08	2.08	1.25
TRINITY_DN41457_c0_g1	APX1	L-ascorbate peroxidase 1	1.83	-	3.37
TRINITY_DN17896_c0_g1	GPX7	Glutathione peroxidase 7	-	-	2.74
TRINITY_DN17462_c0_g2	GSTL3	Glutathione S-transferase L3	-	-	−4.83
TRINITY_DN8736_c0_g1	GSTU19	Glutathione S-transferase U19	-	-	1.31
TRINITY_DN18940_c0_g1	GSTU20	Glutathione S-transferase U20	1.72	7.39	2.26
TRINITY_DN41198_c0_g1	DHAR3	Glutathione S-transferase DHAR3	-	-	1.53

Note: “-” in this table means that the gene was not significantly expressed (*p* > 0.05).

**Table 4 foods-13-01132-t004:** Expression of DEGs related to MAPK signaling pathway in blueberry fruit.

Gene ID	Name	Functional Annotation	log_2_Fold Change		
			CP0d-vs-CK0d	CP2d-vs-CK2d	CP8d-vs-CK8d
TRINITY_DN12758_c0_g1	CAT2	Catalase isozyme 2	1.19	5.95	1.55
TRINITY_DN41456_c0_g1	CAT1	Catalase isozyme 2	−1.71	6.18	-
TRINITY_DN16006_c0_g1	PYL	Abscisic acid receptor PYR/PYL family	-	-	−4.13
TRINITY_DN22372_c0_g1	FLS2	Serine/threonine protein kinase	1.35	-	-
TRINITY_DN15561_c0_g2	NDPK2	Nucleoside diphosphate kinase-like	4.44	-	-
TRINITY_DN18128_c0_g2	NME2	Nucleoside diphosphate kinase	-	-	−3.42
TRINITY_DN25013_c0_g1	WRKY75	WRKY transcription factor 33	-	-	1.25
TRINITY_DN25539_c0_g1	WRKY7	WRKY transcription factor 22	-	-	1.16
TRINITY_DN22163_c0_g1	PR1	Pathogenesis-related protein PR-1 type-like	-	-	2.37
TRINITY_DN30325_c0_g1	NPL7	Transcription factor MYC2	1.51	-	-
TRINITY_DN15873_c0_g1	-	Endo chitinase	-	-	−3.39
TRINITY_DN17764_c0_g1	My16	Calmodulin	2.02	-	-
TRINITY_DN18948_c0_g2	PCM5	Calmodulin-5	-	-	4.42
TRINITY_DN18948_c0_g5	CAM1-1	Calmodulin-1	-	-	7.54

Note: “-” in this table means that the gene was not significantly expressed (*p* > 0.05).

## Data Availability

The original contributions presented in the study are included in the article, further inquiries can be directed to the corresponding author.

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
