# Peer review of "Assessing the Effect of Cold Plasma on the Softening of Postharvest Blueberries through Reactive Oxygen Species Metabolism Using Transcriptomic Analysis"

_foods, 2024, doi:10.3390/foods13071132_

Round 1

Reviewer 1 Report

Comments and Suggestions for Authors

Author Response

Please see the attached file for the comment responses.

Reviewer 2 Report

Comments and Suggestions for Authors

Submitted manuscript presents the new results about the postharvest proceeding or treatment of blueberries by cold plasma generated reactive particles. Significantly higher firmness of treated berries was observed during storage. The methodology is well designed, described experiments and results are clear. Presented conclusions are discussed and supported by the results. Due to my expertise, I have only several minor following comments to improve the text:

- There is the description of plasma generating apparatus only, however, without the description or characteristics of the produced plasma. Please provide details about the generated plasma, as e.g. emission spectra, active particles concentrations to better characteristics of possible reactive particles inducing changes in berries. May be, use some reference to other study describing this apparatus.

- The firmness test is not clear, please specify in more details how the firmness was measured. Figure may be good choice or some reference to other study or technical norm.

Comments on the Quality of English Language

I consider the English as appropriate, however some minor corrections may improve the language.

Author Response

(The authors gave the same response as above.)

Reviewer 3 Report

Comments and Suggestions for Authors

General opinion:

The structure of the article and the ratio of the chapters are appropriate.

·      In my opinion, the material and method chapter needs to be supplemented, the reference to the methods is missing.

·        At the end of the introduction chapter, the wording of the objective is a bit rough, a more precise wording is needed, specifically what were the objectives of the study.

·        In the statistical evaluating of the results chapter, it would be good to write some specific p values in the text.

·        It would greatly improve the evaluation of the results if the authors also revealed a correlation between certain measured parameters.

Other:

·        Page 3, line 123: “Fully mature blueberries (Vaccinium spp. cv. Emerald) were transported from a local blueberry plantation

 How did the authors determine the ripeness of blueberries?

Author Response

(The authors gave the same response as above.)
